# Attitudes, perceptions, and preferences towards SARS CoV-2 testing and vaccination among African American and Hispanic public housing residents, New York City: 2020–2021

Chigozirim Izeogu[1]⊙*, Emily Gill[2]⊙, Kaitlyn Van Allen[2‡], Natasha Williams[2‡], Lorna E. Thorpe[2‡], Donna Shelley[3‡]

1 Department of Neurology, University of Texas Health Science Center at Houston, Houston, Texas, United States of America, 2 Department of Population Health, New York University Grossman School of Medicine, New York, New York, United States of America, 3 Department of Public Health Policy and Management, New York University School of Global Public Health, New York, New York, United States of America

⊙ These authors contributed equally to this work.
‡ KVA, NW, LET and DS also contributed equally to this work.
* Chigozirim.Izeogu@uth.tmc.edu

**Data Availability Statement:** All relevant data are within the paper and its Supporting information files.

## Abstract

### Background

African American and Hispanic populations have been affected disproportionately by COVID-19. Reasons are multifactorial and include social and structural determinants of health. During the onset and height of the pandemic, evidence suggested decreased access to SARS CoV-2 testing. In 2020, the National Institutes of Health launched the Rapid Acceleration of Diagnostics (RADx)- Underserved Populations initiative to improve SARS CoV-2 testing in underserved communities. In this study, we explored attitudes, experiences, and barriers to SARS CoV-2 testing and vaccination among New York City public housing residents.

### Methods

Between December 2020 and March 2021, we conducted 9 virtual focus groups among 36 low-income minority residents living in New York City public housing.

### Results

Among residents reporting a prior SARS CoV-2 test, main reasons for testing were to prepare for a medical procedure or because of a high-risk exposure. Barriers to testing included fear of discomfort from the nasal swab, fear of exposure to COVID-19 while traveling to get tested, concerns about the consequences of testing positive and the belief that testing was not necessary. Residents reported a mistrust of information sources and the health care system in general; they depended more on "word of mouth" for information. The major barrier to vaccination was lack of trust in vaccine safety. Residents endorsed more convenient testing, onsite testing at residential buildings, and home self-test kits. Residents also

**Funding:** NW, DS, and LT received financial support for the research, authorship, and/or publication of this article from the National Institutes of Health (Grant No: 3R01CA220591-03S1 (https://www.nih.gov/). The funders had no role in study design, data collection and analysis, decision to publish, or preparation of the manuscript. The content is solely the responsibility of the authors and does not necessarily represent the official views of the National Institutes of Health.

**Competing interests:** The authors have declared that no competing interests exist.

emphasized the need for language-concordant information sharing and for information to come from "people who look like [them] and come from the same background as [them]".

## Conclusions

Barriers to SARS CoV-2 testing and vaccination centered on themes of a lack of accurate information, fear, mistrust, safety, and convenience. Resident-endorsed strategies to increase testing include making testing easier to access either through home or onsite testing locations. Education and information sharing by trusted members of the community are important tools to combat misinformation and build trust.

## Introduction

The COVID-19 pandemic has had disparate effects on African American and Hispanic populations compared to non-Hispanic Whites. The reasons are multifactorial, involving social and structural determinants of health including differential access to health insurance, high quality medical care, economic opportunities and quality housing [1–3]. During the early stages and height of the pandemic, low-income, predominantly minority populations had less access to SARS CoV-2 testing compared to more affluent, predominantly White populations [4–6]. Testing infrastructure has since improved, however disparities related to SARS CoV-2 testing persist [4, 6, 7]. Testing is a central strategy for curbing the pandemic. Testing provides an opportunity for early diagnosis, monitoring, treatment, and prevention of disease spread. Testing, particularly at the start of the pandemic, was critical for informing policy decisions, including resource allocation and when and where to enact public restrictions such as mask wearing mandates and school and business closings. Even as infection rates have declined, ensuring equitable access to testing remains a high public health priority.

In New York City, patterns of disproportionate COVID-19-related morbidity and mortality mirror those documented nationally. Neighborhoods with higher rates of poverty and densities of Black and Hispanic populations have experienced significantly higher rates of hospitalizations and deaths due to COVID-19 [8]. This is particularly true among those living in public housing. The New York City Public Housing Authority (NYCHA) is the largest public housing authority in the U.S. with more than 400,000 residents. Developments are spread throughout the city but concentrate in lower-income neighborhoods. Median family income is $20,000, and approximately 90% of NYCHA residents are Black or Hispanic. Prior studies have documented that NYCHA residents have a higher burden of conditions associated with poor COVID-19 outcomes relative to those who are White or are living in higher income households [9, 10]. The literature on COVID-19 testing indicates that these disparities are related to a combination of potentially modifiable individual (e.g., health literacy, health status, trust in healthcare system) and structural barriers (e.g., language barriers, wait times at test sites, test properties, cost of isolation) [11–13]. Additionally, prior research examining the attitudes about COVID-19 testing among Black adults found that there was a lack of enthusiasm for testing if individuals were asymptomatic. However, if there was perceived disease susceptibility, such as an exposure or exhibiting pathognomonic symptoms, SARS CoV-2 testing was found to be acceptable [14].

In response to nationwide COVID testing disparities, the National Institutes of Health launched the *Rapid Acceleration of Diagnostics (RADx)- Underserved Populations* initiative to improve SARS CoV-2 testing in underserved communities. In September 2020, New York

University (NYU) Schools of Medicine and Global Public Health received RADx-UP funding to launch the New York City Housing Authority (NYCHA) Resident COVID Response (RCR) project with the primary goal of improving SARS CoV-2 testing among NYC public housing residents. The purpose of this paper is to present findings from a series of focus groups that were conducted to gain a deeper understanding of the barriers to testing among NYCHA residents prior to launching a randomized controlled trial that would compare interventions to increase testing rates in this population. During the course of the study, the FDA gave emergency use authorization to two mRNA COVID-19 vaccines, and in response to significant interest among participants a parallel set of questions exploring similar themes related to SARS CoV-2 vaccination were explored.

## Methods

### Study setting

We identified three neighborhoods in NYC (Harlem, Lower East Side, and East New York) with high densities of public housing residents, a population selected due to their disproportionately high rates of SARS-CoV-2 infection per data released by the New York City Health Department. Prior to implementing the trial, in collaboration with community-based organizations (CBOs) serving these neighborhoods, we conducted a formative assessment to gather additional data on factors that may influence testing uptake and guide adaptations to the proposed interventions operating in each of the distinct neighborhoods. The CBOs, all with a history of impacting the health and wellbeing of NYCHA residents, included Harlem Congregations for Community Improvement, Inc. (HCCI), Henry Street Settlement (HSS), and Church Avenue Merchant Block Association (CAMBA). Together with these CBO partners, we formed the NYCHA Resident COVID Response (NYCHA RCR) Community Steering Committee (CSC), consisting of municipal agencies such as the NYC Health Department and NYC Test and Trace, along with NYCHA leadership and residents. The purpose of the CSC was to foster meaningful collaboration and to address long-standing concerns around mistrust and inequities of resources and power that have informed the dynamic between underserved populations and research or medical entities [15]. Though the CSC was formed for the purpose of the study, members have found it to be a valuable source of bidirectional learning and resource sharing, and agreed to continue meeting beyond the study's conclusion [16]. The study was conducted December 2020 through March 2021, approximately one year after the beginning of the COVID-19 pandemic and after the FDA gave emergency use authorization to two mRNA COVID-19 vaccines.

### Participant recruitment, eligibility and enrollment

We conducted nine virtual focus groups from December 2020 through March 2021 with adult NYCHA residents living in select public housing developments in Brooklyn and Manhattan. Eight focus groups were conducted in English, and one was conducted in Spanish. Participants were considered eligible if they were 18 years and older and lived in NYCHA. The NYCHA RCR CSC provided feedback on the focus group guide before it was finalized. Residents were recruited from six NYCHA housing developments through flyers distributed in buildings and on-site at CBO locations. CBOs also promoted the study at public and virtual events. Residents were encouraged to contact the study research assistant to obtain additional information. Once contacted, the research team screened individuals for eligibility and provided a date and time for the virtual meeting. Because of the low-risk nature of the study, in addition to the fact that we were unable to convene in-person, we were approved to collect verbal consent from participants. Participants were read the consent form over the phone, and their approval was

documented in a secure electronic database. Thirty-six residents participated in the focus groups using Zoom. To maintain anonymity and confidentiality, upon logging on each participant was renamed with an alias. Focus groups were recorded and then transcribed verbatim for analysis. Focus groups lasted approximately one hour, and participants were reimbursed with a $30 Amazon electronic gift card via email. The study protocol was approved by the Institutional Review Board of NYU Grossman School of Medicine. (i20-01636).

### Focus group guide

The guide was informed by the integrated behavior model, which posits that attitudes, perceived norms and personal agency (i.e., self-efficacy and perceived control) are associated with intention to modify behavior [15]. The model further suggests that the salience of the behavior and environmental constraints can further modify behavior. Interview questions assessed knowledge about and attitudes towards testing and vaccination, including confidence in accuracy of the tests, perceived community norms related to testing and vaccination, risk perception, and environmental constraints related to testing and vaccination. We also explored how residents accessed information about COVID-19 [17, 18]. The study was conducted December 2020 through March 2021, approximately one year after the beginning of the COVID-19 pandemic and after the FDA gave emergency use authorization to two mRNA COVID-19 vaccines. Therefore, we included a parallel set of questions exploring similar themes related to vaccination.

### Data analysis

Focus group findings were analyzed using rapid qualitative methods [19]. This approach is useful when conducting formative research in which there is a time limited engagement and the goal is to conduct a deductive analysis. We assigned a code for each of the focus group guide questions (i.e., themes) to create a template to guide the systematic extraction and analysis. Four members of the research team engaged in an iterative process of transcript review and coding that did allow for additional codes to emerge. Coders used the template to document main concepts and illustrative quotes. the analysis began with the team reviewing two transcripts and meeting to review side-by-side comparisons and to address discrepancies. This process was repeated using two more transcripts and the revised template. Once the template was finalized, two members of the team completed the analysis and created a final matrix of findings that were then synthesized across themes.

## Results

We present main themes and subthemes that emerged from the analysis. Themes are grouped into reasons for and against getting tested, testing preferences and strategies to increase testing, as well as attitudes towards vaccination against SARS CoV-2. Mistrust emerged as an overarching theme across the majority of the focus group discussions.

### Why residents were not getting tested

**Fear of testing.**   Residents expressed fears about storage of and access to test results, with some reluctant to provide personal information. One resident (FG2) noted, *"some may not want to give their personal information or to feel that they are being traced, so they hold back altogether on being tested."* Some also feared the test itself, with some citing concerns that the test itself might transmit SARS CoV-2. One resident (FG1) stated, *"Black folk and poor folk and folk of color, we're not very trusting of our government. . .I had a woman say to me. . ."How*

*do I know they're not givin' it to me when they swab my throat or my nose?"* Others reported that the nasopharyngeal swab test was painful, or that they witnessed others experiencing discomfort. Residents were also concerned about the potential consequences of a positive test. This included being forced to isolate themselves from their families without their consent. As one resident (FG4) said *"not everybody understands quarantine,"* explaining that some people have the conception that it's *"you being locked up in a room, and the key is thrown away with nobody else around you."*

## Misinformation

Residents received information about COVID-19 from a wide range of sources, including the news, local politicians, and word of mouth. Word of mouth emerged as a primary source of misinformation. Participants reported that many residents did not believe that SARS CoV-2 was real but rather a government tactic to scare racial/ethnic minority groups. As one resident (FG3) said, *"many people feel that Trump is just doing this because he wants to keep us down. He doesn't want us to work anymore, he's trying to scare us."* Some residents (FG5, FG1) heard from neighbors that the novel coronavirus was *"fake,"* and not as serious as it has been depicted in the media: *"a lot of people, they just don't know what's going on. . .They don't believe in COVID, or that it's real."* Several residents observed that young people *"seem to be unaware of the severity of COVID. It's almost like the younger you are, you think you're [impervious]. You're Superman, so COVID won't affect you"* (FG9). This was described in the context of concerns about adolescents not wearing masks in the elevators.

## Challenges getting tested

For those who were tested or tried to get tested, one of the main challenges was having to wait in long lines and the perceived associated risk of exposure to infection while waiting. As one resident (FG3) said, *"the lines are long, and we can catch COVID [while waiting in line]."* Similarly, residents who reported having to travel on public transportation to get to a testing location were concerned about getting infected with SARS CoV-2. *"We worry to go on the bus because if we go on the bus, we are worried to catch COVID and then worried to take the train because we could catch COVID"* (FG3). Another potential barrier was the possibility of paying for a test. A few participants described receiving a bill despite having Medicaid, which covers testing. One resident (FG1) recalled, *"Some people went. . .and they got a big bill from being tested for the coronavirus. . .a lot of people don't want to go to these sites because [they] are getting billed for the corona test."*

## Why residents were getting tested

More than half of participants had been tested at least once. Reasons for getting a SARS CoV-2 test included: preparing for a medical procedure, exposure to someone with the virus and having a high-risk job. As one resident (FG3) noted, *"I'm always in the community. I volunteer in the community."*

## Preferences for testing

Most residents preferred getting tested in a clinical setting. However, they also endorsed onsite testing options (i.e., in their housing development), with the caveat that those conducting the testing had some clinical training. One resident (FG5) proposed the community centers that are on-site at all housing developments, saying *"as far as NYCHA is concerned, I think they should have [testing] in the nearest hospital by their facility or even within their community*

center. *Every development has a community center. . .Everybody knows where the community center is, and it's easy to access.*" Most residents were open to self-administration, with guidance. Residents preferred testing methods that caused less discomfort and were interested in home test kits. One resident (FG4) described the ideal test as one that was similar to an at-home pregnancy test, *"if it's a testing kit as a woman would use to find out if she's going to have a baby, I'm all for it."* Some residents did express concerns about their ability to conduct a self-test correctly. For example, one resident (FG9) said, *"I feel doing it yourself, I would be wary about whether or not it was done correctly. That's why I would feel I would rather have someone who is educated in doing do it, rather than me at home trying to do it myself."*

## Strategies for increasing testing

Resident recommendations for increasing the uptake of testing in the community are summarized in Table 1. Residents suggested disseminating information in multiple languages, saying (FG3) *"keep constant communication, especially to those who English is a second language. Messages should be in their home language."* They also described setting up information tables in the housing developments, sending information to residents' homes, and communicating through flyers, social media posts and emails. As one resident (FG3) suggested, *"use social media for young people and door to door for older people."* Residents also responded positively to the idea of a community health worker (CHW) model in which CHWs were available to provide this information and other resources. Residents were particularly intrigued by the idea of a saliva test, saying, *"it's convenient and less painful"* (FG6).

All of the residents emphasized that messages must come from a trusted source, and preferably someone from within their community. One resident (FG4) noted, "*The source of information should be. . . someone who looks like us, who talks like us."* They liked the idea of one-on-one interactions with a knowledgeable person from the community: *"that's a great thing, [it] just has to be community structured. Meaning it has to be discrete, very one-on-one"* (FG4).

## Attitudes towards vaccines

Focus groups were conducted shortly after the first vaccine in the U.S. received its emergency use authorization and availability was more limited. At that time, most residents adopted a wait and see approach and were not planning to get vaccinated. Residents raised concerns about safety related to the perceived speed with which the vaccination was brought to market, saying *"it hasn't been fully tried. It hasn't been fully tested. This is a cross your fingers and wish us luck type of thing"* (FG5). As one resident (FG3) explained, *"I want to hear experiences from the people and not just one person. I'm going to watch the news, of course, and I want to hear people like the nurses that got it and politicians."* Residents also expressed the need for

**Table 1.  Resident suggestions for increasing acceptability and uptake of SARS CoV-2 testing.**

| |
|---|
| Provide accurate easy to read information (e.g., develop a COVID handbook) |
| Host events/testing in housing developments or community centers |
| Create multilingual materials |
| Continue conducting outreach |
| Knock on doors with COVID tests on hand |
| Post to social media/email information |
| Create age-appropriate materials |
| Make self-tests available |
| Keep Testing Convenient |

information about the safety and effectiveness among different populations including racial/ethnic minorities, older adults, and those with chronic health conditions. They (FG2, FG6) were not convinced that these groups were appropriately represented in the clinical trials, saying *"when they first started the trial runs on the vaccine. . . they were not getting that many people of color to be participants in the research"* and *"I feel like they've only tested those vaccines on fairly healthy people. I don't know how it would affect me with my whole transplant issue, with all the medications that I'm taking."* These beliefs persisted even in the final focus group, which occurred after vaccines became more widely available. The few residents willing to get the vaccine acknowledged that they did not know the long-term effects and indicated that they would still have to protect themselves with a mask and maintain social distance. One resident (FG3) who wanted to get vaccinated was rethinking this decision based on others' opinions, *"[I] think I'm going to take it. I really don't have doubts, but people put doubts in my head sometimes."*

In the English language focus groups that included primarily African American participants, many residents described a general lack of trust in the government: *"do not trust them. Do not trust the government"* (FG4). This lack of trust extended to the scientific community. The Tuskegee experiment was specifically mentioned as an example of a history of unethical treatment of Black Americans. As one resident (FG4) stated, the *"That Tuskegee experiment has damaged the government badly."* Another noted that the *"history of things regarding shots and vaccines is where the stress value lies."*

In contrast, those in the Spanish language group reported no vaccine hesitancy except for some concerns about having a preexisting condition that might result in a side effect from the injection. For these participants (FG8), the range of trusted sources of information on the vaccine was quite broad, and included *"the people that make the vaccine, the company, the doctors, the scientists,"* healthcare staff, and *"university students doing the research."*

## Discussion

Among a minority, low-income population of residents living in multiunit public housing developments, about half of the participants had never obtained a SARS CoV-2 test. Similarly, in focus groups conducted after the vaccine was available most participants had no plans to get vaccinated. Reasons for testing were largely driven by the need to prepare for a medical procedure, work-related requirements, or a known exposure to someone who tested positive.

We identified a range of barriers to SARS CoV-2 testing that could be addressed with increased engagement and communication from trusted sources to reduce confusion and concerns about the purpose and safety of testing and to reduce challenges related to accessing testing. General mistrust in the information they were receiving was particularly strong among African American participants. Our findings were consistent with a recent study examining SARS CoV-2 testing perceptions in under-resourced African American neighborhoods in urban and rural Alabama [20]. In that study, Bateman and colleagues reported that poor access, cost, and fear of contracting SARS CoV-2 through the act of testing itself were predisposing barriers to testing [20]. Our findings were also highly comparable with a literature review of COVID testing hesitancy, in which barriers were organized according to the "three delays" model: planning (individual factors such as knowledge about testing), process (characteristics of the test itself, such as accessibility of testing sites) and outcomes (consequences of the COVID-19 testing results, such as consequences on employment) [11].

Mistrust emerged as a major underlying theme throughout the focus groups and ranged from mistrust of government leaders, the news media and general mistrust of research and the health care system. Conspiracy theories and misconceptions about the origin and nature of

various diseases have been described among minority populations prior to the COVID-19 pandemic [21, 22]. For example, among a nationally representative e-mail survey of 868 African Americans aged 18–50 years (February–April 2016) substantial percentages agreed that HIV is man-made (31%) and that the government is withholding a cure for HIV (40%) [23]. These types of theories and myths can affect adherence to treatment as well as disease screening and surveillance programs [24–26].

The historic context of medical mistrust among the African American community came up in the form of reference to the Tuskegee Syphilis Study, a U.S. Public Health Service run research study in which Black men were enrolled without their consent and intentionally left without treatment when one was available [27]. Similar to Bateman and colleagues' study in which participants brought up the Tuskegee Syphilis Study in three of their focus groups, several Black participants referenced the study in a few of our focus groups [20]. It is notable, that the African American NYCHA residents still share in the collective trauma from the memory of the unethical treatment of Blacks in this study and link it to their decision-making regarding testing and vaccination during the COVID-19 pandemic. Moreover, as Willis et. al., mention in their study, distrust of the medical establishment by Black Americans goes well beyond just the one incident, and that "racism within the medical establishment is ongoing, and Black/African Americans do not need an extensive knowledge of the history of medical racism to inform their view of vaccines when many only need to consider recent experiences" [28].

Spanish-speaking participants appeared less concerned about vaccine safety and were less likely to question the effectiveness of the vaccine. In general, they were more willing to get vaccinated compared with the majority of African American English-speaking participants. These findings are consistent with published survey results from a national survey conducted Sept. 1–15, 2020, in which 18% of Black Americans and 40% of Hispanics reported trusting the efficacy of the COVID-19 vaccine; fewer than half of Black Americans intended to get vaccinated against COVID-19 [29]. In this study, the Spanish-speaking participants' greater willingness to be tested may have been due to timing. This focus group was hosted later in the timeline of vaccine distribution and therefore residents would have been exposed to more information that may have impacted their beliefs. Another possibility for differing attitudes between Black and Hispanic participants may be a shorter history of experiencing injustices in the US health care system due to length of time in the country.

Residents suggested several strategies to increase testing that directly address the barriers they described. This included more tailored and targeted communication that is delivered by trusted sources. Clear, accurate and consistent messaging will continue to be important as the pandemic and associated changes in guidelines for testing and vaccination change. Similarly, the overwhelming amount of information available about the vaccine and news about variants has created even more confusion in the community about whether or not the current vaccines work. Again, ongoing communication, using platforms that offer language concordant information and reach a range of age groups and communities, is needed to reinforce the benefits of COVID-19 testing and the need for vaccination. Residents requested that vaccine efficacy and safety data be reported by subgroups (e.g., older adults, racial/ethnic minorities). They also endorsed patient navigator and community health worker models that provide opportunities to work with trained individuals who know the community and are viewed as a trusted source of information about COVID-19.

Our study had several limitations. First, we did not collect demographic data other than race/ethnicity. This was in response to our CBO partner's concerns that residents would view this as too intrusive and, therefore, they would be less likely to engage in the study. Second, we did try to conduct an equal number of English and Spanish language focus groups, however this population was reluctant to enroll, therefore we only conducted one in Spanish. Finally,

there is a chance that our study was subject to voluntary response bias, as our participants were self-selected volunteers. This could have particular relevance to the topic of mistrust of the medical establishment; our recruitment materials clearly indicated the focus groups were hosted by New York University.

This study is one of the first few studies exploring the attitudes and beliefs about COVID-19 testing and vaccines among Black and Hispanic public housing residents, a population that has disproportionately experienced negative outcomes of the COVID-19 pandemic. Similar to others, we found that barriers to SARS CoV-2 testing and vaccination centered on themes of a lack of accurate information, fear, mistrust, safety, and convenience. Resident-endorsed strategies to increase testing included making testing easier to access either through home or onsite testing locations. In addition to increasing access, providing culturally appropriate education and information sharing by trusted members of the community were identified as vital tools to combat misinformation and build trust. Closing the racial/ethnic and economic gaps in testing and vaccination will require continued engagement with communities to provide data and other information that can change attitudes and beliefs that are aligned with vaccination. With the increasing availability of at-home test kits, future research should investigate accessibility, usage, and how attitudes and beliefs continue to evolve about this issue along with other COVID preventive measures.

## Supporting information

**S1 File.**
(DOCX)

## Acknowledgments

We would like to thank our colleagues at the New York City Housing Authority and partnering community-based organizations (Harlem Congregations for Community Improvement, Inc., Henry Street Settlement, and Church Avenue Merchant Block Association) for their assistance in conducting this research.

## Author Contributions

**Conceptualization:** Natasha Williams, Lorna E. Thorpe, Donna Shelley.

**Formal analysis:** Chigozirim Izeogu, Emily Gill, Kaitlyn Van Allen, Donna Shelley.

**Funding acquisition:** Natasha Williams, Lorna E. Thorpe, Donna Shelley.

**Investigation:** Donna Shelley.

**Methodology:** Donna Shelley.

**Project administration:** Emily Gill, Kaitlyn Van Allen.

**Supervision:** Natasha Williams, Lorna E. Thorpe, Donna Shelley.

**Writing – original draft:** Chigozirim Izeogu.

**Writing – review & editing:** Chigozirim Izeogu, Emily Gill, Kaitlyn Van Allen, Natasha Williams, Lorna E. Thorpe, Donna Shelley.

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
