## [Decision Letter · Decision Letter 0]

18 Apr 2022

PONE-D-21-34989Attitudes, perceptions, and preferences towards SARS Cov-2 testing and vaccination among African American and Hispanic public housing residents, New York City: 2020-2021PLOS ONE

Dear Dr. Shelley,

Thank you for submitting your manuscript to PLOS ONE. After careful consideration, we feel that it has merit but does not fully meet PLOS ONE’s publication criteria as it currently stands. Therefore, we invite you to submit a revised version of the manuscript that addresses the points raised during the review process.

We look forward to receiving your revised manuscript.

Kind regards,

Sze Yan Liu, PhD

Academic Editor

PLOS ONE

Journal Requirements:

Additional Editor Comments:

The reviewers and I agree on the importance of this topic. However, the reviewers bring up multiple good points about the need for more detail and greater clarity in your study that we would like to see addressed.

Reviewers' comments:

Reviewer's Responses to Questions

**Comments to the Author**

1. Is the manuscript technically sound, and do the data support the conclusions?

Reviewer #1: Yes

Reviewer #2: Yes

Reviewer #3: No

2. Has the statistical analysis been performed appropriately and rigorously? 

Reviewer #1: N/A

Reviewer #2: N/A

Reviewer #3: N/A

3. Have the authors made all data underlying the findings in their manuscript fully available?

Reviewer #1: No

Reviewer #2: Yes

Reviewer #3: No

4. Is the manuscript presented in an intelligible fashion and written in standard English?

Reviewer #1: Yes

Reviewer #2: Yes

Reviewer #3: No

5. Review Comments to the Author

Reviewer #1: Thank you for the opportunity to review an interesting paper with significant issues identified for increasing uptake of testing and vaccination of African American and Hispanic people.

I do think that the manuscript would benefit from some changes and further considerations, which I outline for each section.

Introduction

The Introduction described the study well and what was done, as well as some brief consideration of the issues faced regarding testing with African American and Hispanic people. What is needed is some more consideration of the literature regarding the topics of testing and vaccinations - the Introduction is currently very focused on the project, rather than what literature informed the project. Some discussion of what we know from literature about the determinants/predictors of willing to be tested (and vaccinated) is needed. There may not be work with these specific populations, but there should be some work that has been done more generally.

The Introduction should also end with a purpose statement. Rather than focusing on what was done (the focus groups), the purpose of the study - e.g. to examine African American and Hispanic people's perceptions of COVID testing and vaccinations, and barriers/enablers to both - would situate the paper more as a research-oriented paper.

I did question whether the information on the project should move to the Method. I am happy for it to stay in the Introduction, but more research literature or theory is needed in the Introduction.

Method

The Method was described well. More information is needed regarding how participants were recruited - for example, how were they recruited (e.g. social media) and how did they get in touch with the researchers.

Some example interview questions should be included in the Method.

The analysis was said to use rapid qualitative methods. Some more information on the use of this approach and a justification is needed. I wondered why an approach such as thematic analysis was not utilised. It is also stated that a deductive approach was used - however, without theory or models and previous literature presented in the Introduction, it is difficult to know how this was done. The integrated behaviour model is mentioned in passing in the Method, but if this was the underlying approach, this needs to be discussed in the Introduction.

Results

The results provide interesting data. I did think some more nuance in discussion of results was needed - at times, statements are made that make it sound like every participant was in agreement. Consider some more discussion of to what extent certain attitudes were prevalent.

Some themes received very little attention - the theme 'Why residents are getting tested' was very short. At other times, statements are made very briefly - e.g concerns about getting a positive tests are given one sentence only.

Focus group numbers and participant numbers should be included so the reader can see the diversity in use of participant responses.

It is mentioned towards the end of the Results that there was a difference between African American and Hispanic responses. This perhaps should be mentioned earlier, if there was such a marked difference.

I did wonder if both attitudes to testing and attitudes to vaccines could be discussed together in each theme - e.g. the misinformation theme could include reflections regarding both - but I will leave this up to the authors to decide if they want to do this.

Again, I would have liked to see, given the deductive approach, how it reflected parts of the models used. The models do not need to be explicitly mentioned in the Results, but there should be some connection between their description in the Introduction and then the themes in the Results.

Discussion

Discussion was well done. I would like to see more reference to COVID literature - again, this may not necessarily be from studies with African Americans and Hispanic people (if there are only a few), but issues such as attitudes to vaccines, questioning existence of the virus etc., have been investigated and the current results should be examined in light of these findings.

Data availability

I wasn't sure if the transcripts need to be potentially available to other researchers (e.g. on request). I will let the Editor decide that as I am very familiar with the data policies of the journal.

Thank you and I hope the review is helpful to the authors.

Reviewer #2: Authors present themes in the perceptions of African-American and Hispanic-American individuals living in New York public housing units on COVID-19 testing and vaccination access and uptake. This article will be helpful in informing future COVID-19 testing and vaccination initiatives aiming to meet the needs of low-income and racial minority populations in this region. Minor revisions are suggested.

1. Introduction 69-70 highlights broad applications of the findings of this study. Were they also directly used to improve testing services by RADx, NYU, or other institutions or initiatives noted in the introduction?

2. Introduction. Provide context on where the nation was in terms of testing and vaccination availability at the time that this intervention was carried out. Were opinions shared on a real or hypothetical vaccine candidate? A bit on this is discussed in results but it should be discussed here.

3. Introduction 59-67. This description of the focus group discussion design is likely more appropriate for the methods section. It would be helpful to include an explanation (including references) in the introduction on the importance of working with CBOs such as those in the design of such an intervention.

4. Introduction 63 "Collectively, they designed and conducted..." recommend rewording to clarify if "they" includes the authors/NYU because as it stands this seems to suggest this is referring to the CBOs.

5. Methods 87-90. Please provide examples of focus group questions.

6. Methods. 92-94. Clarify what the authors "protocol-driven, deductive" approach was. What protocol or theory was informing the analysis? The referenced article does not add clarity to the statement. Were the domain names identified based on a particular guiding framework? Based on consultation with the CSC?

7. Results. Demographics. Was any participant demographic information beyond race collected that might help contextualize the study findings? Age, for example? In results 194-203 you seem to suggest most of the 8 focus groups were with African-Americans and 1 was with Hispanic Americans. It would be helpful to clarify the amount of participants identifying as African-American and/or Hispanic American included in the sample.

8. Results. 149. Suggest "development" -> "housing development"

9. Results. Table 1. Would be helpful to include how often these suggestions came up to illustrate how well they are endorsed by the sample. Did only one person in one focus group suggest this? Did it come up in each of the 9 focus groups?

10. Discussion. 210-214. You describe differences with respect to some of these conclusions among Af-AM and His-Am participants in the results section- it will be important to preserve these distinctions in this section so as to not conflate the perceptions and experiences of these two groups inappropriately.

11. Discussion 249-262. Some of this is new data that should be in the results, likely in the section titled "Strategies for increasing testing," not introduced for the first time in the discussion. Eg. "with the vaccine now widely available, residents who are vaccinated were confused about the indications for future testing."

Reviewer #3: The authors describe a qualitative focus group study to understand attitudes towards SARS-CoV-2 testing and vaccination to inform an intervention to increase testing among public housing residents. The topic is important and interesting, but the manuscript is lacking in its framing in the introduction, the description of methods, and valuable interpretation/contextualization in the discussion. It also has significant readability issues. Specific comments for each section follow:

Introduction:

- The authors should describe and cite existing literature on perceptions towards testing and vaccination among marginalized groups

- The last sentence of the introduction is the first time that vaccination is mentioned - why did the study also include perceptions towards vaccination? The authors need to provide framing and justification for this

Methods:

- The authors might consider briefly explaining the integrated behavioral model

- Many more details are needed on the specific interview questions and topics covered in the focus groups

- More information on the CBO-academic partnerships would be of interest and relevance to other researchers

Results:

- The authors should consider first describing the demographic composition of their focus group participants

- A lot of the wording in this section is awkward/clunky, i.e. "reasons for getting or not getting tested"

- "testing data storage and access" reads in a confusing way, the authors might instead consider "storage of and access to test results"

- A lot of the sentences do not use parallel structure, which impedes readability, i.e. "Residents suggested disseminating information through flyers, social media posts and emails, and that the information be translated into multiple languages." and "They also described setting up information tables in the developments, sending information to residents’ homes, and some suggested knocking on doors."

- The authors should avoid interpreting in Results rather than Discussion, ie "This comment demonstrates the potential negative impact of misinformation often spread by word of mouth."

- It is unclear how the authors classified and distinguished fears of testing from misinformation from mistrust, since there seemed to be a lot of overlap across the three

- Why were the responses from the predominantly Black group compared with the one Spanish speaking group, rather than with participants of other races in any group?

Discussion:

- "general lack of knowledge about testing recommendations, a similar lack of knowledge about risk factors for infection," - this was not presented at all in the Results

- The authors should discuss similar interventions that have been effective

- It would be helpful for the authors to suggest more concrete solutions to mitigating mistrust and increasing access (ie transportation, ensuring that tests are free, etc.)

- The authors need a limitations paragraph, especially detailing the potential biases (ie voluntary response, potential underrepresentation/lack of representation of certain perspectives or subgroups), shortcomings of methods, limited generalizability to other populations, etc.

6. PLOS authors have the option to publish the peer review history of their article (what does this mean?). If published, this will include your full peer review and any attached files.

Reviewer #1: No

Reviewer #2: **Yes: **Debbie Dada

Reviewer #3: No

---

## [Author Response · Author response to Decision Letter 0]

20 Oct 2022

October 15, 2022

Dear Dr. Liu,

Thank you for reviewing our manuscript and for the opportunity to revise it to meet PLoS One’s publication standard. Please find listed below the comments submitted by the reviewers and our response to each comment in bolded italicized font. Please let us know if we can clarify anything further.

Reviewer #1: Thank you for the opportunity to review an interesting paper with significant issues identified for increasing uptake of testing and vaccination of African American and Hispanic people. I do think that the manuscript would benefit from some changes and further considerations, which I outline for each section.

Introduction

The Introduction described the study well and what was done, as well as some brief consideration of the issues faced regarding testing with African American and Hispanic people. What is needed is some more consideration of the literature regarding the topics of testing and vaccinations - the Introduction is currently very focused on the project, rather than what literature informed the project. Some discussion of what we know from literature about the determinants/predictors of willing to be tested (and vaccinated) is needed. There may not be work with these specific populations, but there should be some work that has been done more generally.

RESPONSE. We updated the introduction to address the reviewer’s comments which included reviewing additional literature and moving sections that described the methods to that section. We agree with this reviewer’s comment that much of what was previously included in the Introduction is more fitting for the Methods section.

REVIEWER. The Introduction should also end with a purpose statement. Rather than focusing on what was done (the focus groups), the purpose of the study - e.g., to examine African American and Hispanic people's perceptions of COVID testing and vaccinations, and barriers/enablers to both - would situate the paper more as a research-oriented paper.

REPONSE. We thank the reviewer for this comment and have added a purpose statement at the end of the Introduction section. 

REVIEWER. The Method was described well. More information is needed regarding how participants were recruited - for example, how were they recruited and how did they get in touch with the researchers.

RESPONSE. We have added more information including eligibility criteria and more detail on recruitment strategies. 

REVIEWER. Some example interview questions should be included in the Method.

RESPONSE. The interview questions will be provided in the supplemental appendix.

REVIEWER. The analysis was said to use rapid qualitative methods. Some more information on the use of this approach and a justification is needed. I wondered why an approach such as thematic analysis was not utilized. It is also stated that a deductive approach was used - however, without theory or models and previous literature presented in the Introduction, it is difficult to know how this was done. The integrated behavior model is mentioned in passing in the Method, but if this was the underlying approach, this needs to be discussed in the Introduction.

RESPONSE. We have expanded the description of the theoretical framework that informed the focus group guide. Rapid qualitative methods and analysis is a well described approach that emerged from the need, in situations such as in the case of formative work, where the timeline is short and the data is needed to inform modifications to be used in a larger randomized controlled trial. We used the integrated behavioral theory framework to inform the questions that were asked. Then using those questions, we created a template for the deductive analysis. We used a team based, highly iterative process that did allow for additional codes to emerge. This process doesn’t preclude more inductive assessments in the future

REVIEWER. Results. The results provide interesting data. I did think some more nuance in discussion of results was needed - at times, statements are made that make it sound like every participant agreed. Consider some more discussion of to what extent certain attitudes were prevalent. Some themes received very little attention - the theme 'Why residents are getting tested' was very short. At other times, statements are made very briefly - e.g. concerns about getting positive tests are given one sentence only.

RESPONSE. In many of the focus groups, there was agreement among the participants as it pertains to specific topics/themes, particularly among the African American participants. Where there were differences, we did highlight those themes, specifically in the last two paragraphs of the results section. For example, the results indicate that the Spanish language focus group participants did not report vaccine hesitancy and had much higher trust in mainstream information sources. 

REVIEWER. Focus group numbers and participant numbers should be included so the reader can see the diversity in use of participant responses.

RESPONSE. Focus group numbers have been added to participant responses. As part of the transcription process, any distinct or identifying information between participants is removed to ensure the data is completely de-identified. As such, we cannot provide individual participant numbers. 

REVIEWER. It is mentioned towards the end of the Results that there was a difference between African American and Hispanic responses. This perhaps should be mentioned earlier, if there was such a marked difference.

RESPONSE. We tried to recruit equal numbers of Hispanic and African American participants; however, Hispanic individuals were reluctant to enroll. This may have been due to the sociopolitical context of the time in terms of concerns about immigration status. As a result, we only had one Hispanic focus group. Due to the limited amount of data from this population, we highlight some of the clear difference, however, we cannot make any additional conclusions about substantive differences between the two groups. 

REVIEWER. I did wonder if both attitudes to testing and attitudes to vaccines could be discussed together in each theme - e.g. the misinformation theme could include reflections regarding both - but I will leave this up to the authors to decide if they want to do this.

RESPONSE. RADx-UP was and is an initiative to increase SARS-Cov-2 testing. However, once we launched the study, the vaccine was made available so we included information about the vaccinations because participants were interested in discussing it. We agree that it wasn’t explored as robustly as the testing questioning, however, vaccination uptake/hesitation was included in this manuscript because it became a major point of discussion during the focus group sessions given the timing of the study which coincided with the emergency use approval of the first vaccinations against Sars-Cov-2 in the U.S.

REVIEWER. Again, I would have liked to see, given the deductive approach, how it reflected parts of the models used. The models do not need to be explicitly mentioned in the Results, but there should be some connection between their description in the Introduction and then the themes in the Results.

RESPONSE. As per the response above, we have expanded on this section. 

REVIEWER. Discussion was well done. I would like to see more reference to COVID literature - again, this may not necessarily be from studies with African Americans and Hispanic people (if there are only a few), but issues such as attitudes to vaccines, questioning existence of the virus etc., have been investigated and the current results should be examined in light of these findings.

RESPONSE. This is addressed in the updated introduction.

REVIEWER. Data availability. I wasn't sure if the transcripts need to be potentially available to other researchers (e.g. on request). I will let the Editor decide that as I am very familiar with the data policies of the journal.

RESPONSE. The data is available for review upon request.

Reviewer #2: Minor revisions are suggested.

Introduction 69-70 highlights broad applications of the findings of this study. Were they also directly used to improve testing services by RADx, NYU, or other institutions or initiatives noted in the introduction?

RESPONSE. Yes, the study was completed to inform the development of an intervention to increase uptake of Sars-CoV-2 testing among African American and Hispanic residents in public housing. 

REVIEWER. Introduction. Provide context on where the nation was in terms of testing and vaccination availability at the time that this intervention was carried out. Were opinions shared on a real or hypothetical vaccine candidate? A bit on this is discussed in results but it should be discussed here.

RESPONSE. We have provided this information in the Methods section under the “Context” sub-heading. The focus groups began roughly a year after the start of the pandemic, by which point COVID tests were widely available and routine testing encouraged. The bulk of the discussions around vaccine happened after vaccines had been rolled out and approved for emergency usage by the FDA. 

REVIEWER 

Introduction 59-67. This description of the focus group discussion design is likely more appropriate for the methods section. It would be helpful to include an explanation (including references) in the introduction on the importance of working with CBOs such as those in the design of such an intervention.

RESPONSE. We thank the reviewer for this comment and we have moved this portion to the Methods section. 

REVIEWER

Introduction 63 "Collectively, they designed and conducted..." recommend rewording to clarify if "they" includes the authors/NYU because as it stands this seems to suggest this is referring to the CBOs.

RESPONSE. This is addressed in the updated Introduction.

REVIEWER. Methods 87-90. Please provide examples of focus group questions. 

RESPONSE. Focus group interview guide questions available in supplemental appendix 

REVIEWER. Methods. 92-94. Clarify what the authors "protocol-driven, deductive" approach was. What protocol or theory was informing the analysis? The referenced article does not add clarity to the statement. Were the domain names identified based on a particular guiding framework? Based on consultation with the CSC?

RESPONSE. See the previous response to this question. Given the nature of the COVID-19 pandemic and the need to generate actionable next steps to inform the final design of the intervention, the analysis was conducted using a rapid qualitative analysis process. The questions were drawn from integrated behavior model with guidance from the CSC, and the main code names were derived from the questions.

REVIEWER. Results. Demographics. Was any participant demographic information beyond race collected that might help contextualize the study findings? Age, for example? In results 194-203 you seem to suggest most of the 8 focus groups were with African-Americans and 1 was with Hispanic Americans. It would be helpful to clarify the number of participants identifying as African-American and/or Hispanic American included in the sample.

RESULTS. No other demographic information beyond race/ethnicity were collected under suggestion by a CBO partner because of concern that too many personal questions may be considered as too intrusive by the study population. This is a limitation of the study. As for the breakdown of the demographics, of our 36 participants 4 identify as Hispanic/Latino; 30 Black or African American, and 2 Asian. 

REVIEWER. Results. Suggest "development" -> "housing development"

RESPONSE. This is addressed in the updated manuscript.

REVIEWER. Discussion. 210-214. You describe differences with respect to some of these conclusions among Af-AM and His-Am participants in the results section- it will be important to preserve these distinctions in this section so as to not conflate the perceptions and experiences of these two groups inappropriately.

RESPONSE. We have highlighted the thematic distinction with supporting quotes between these two groups in the final two paragraphs of the results section.

REVIEWER. Discussion 249-262. Some of this is new data that should be in the results, likely in the section titled "Strategies for increasing testing," not introduced for the first time in the discussion. Eg. "with the vaccine now widely available, residents who are vaccinated were confused about the indications for future testing."

RESPONSE. We have edited this in the discussion section.

Reviewer #3: The topic is important and interesting, but the manuscript is lacking in its framing in the introduction, the description of methods, and valuable interpretation/contextualization in the discussion. It also has significant readability issues. Specific comments for each section follow: Introduction:- The authors should describe and cite existing literature on perceptions towards testing and vaccination among marginalized groups

RESPONSE. We have added additional references to the literature. 

REVIEWER- The last sentence of the introduction is the first time that vaccination is mentioned - why did the study also include perceptions towards vaccination? The authors need to provide framing and justification for this

RESPONSE. The study was designed specifically to address testing and not vaccination uptake or hesitancy. We have added in the introduction that the vaccine became available in the middle of our formative research and therefore we decided we needed to add those questions to our focus group guide. 

REVIEWER. Methods: The authors might consider briefly explaining the integrated behavioral model. Many more details are needed on the specific interview questions and topics covered in the focus groups.

RESPONSE. This is addressed in the updated Methods. Topics covered are in the focus group interview guide which is available as a supplement. We do outline the key topics in the methods section.

REVIEWER. More information on the CBO-academic partnerships would be of interest and relevance to other researchers

RESPONSE. We have expounded on this partnership in the Methods section under the Study Setting subheading. 

REVIEWER Results:- The authors should consider first describing the demographic composition of their focus group participants

RESPONSE. As previously discussed, no other demographic information beyond race/ethnicity were collected under suggestion by a CBO partner because of concern that too many personal questions may be considered intrusive by the study population. This is a limitation of the study. 

REVIEWER.

- A lot of the wording in this section is awkward/clunky, i.e. "reasons for getting or not getting tested"

- "testing data storage and access" reads in a confusing way, the authors might instead consider "storage of and access to test results" A lot of the sentences do not use parallel structure, which impedes readability, i.e. "Residents suggested disseminating information through flyers, social media posts and emails, and that the information be translated into multiple languages." and "They also described setting up information tables in the developments, sending information to residents’ homes, and some suggested knocking on doors."

RESPONSE. We thank the reviewer for this comment and have revised to improve clarity.

REVIEWER The authors should avoid interpreting in Results rather than Discussion, ie "This comment demonstrates the potential negative impact of misinformation often spread by word of mouth."

RESPONSE. We have removed this commentary from the Results section. 

REVIEWER- It is unclear how the authors classified and distinguished fears of testing from misinformation from mistrust, since there seemed to be a lot of overlap across the three

RESPONSE. These themes are not mutually exclusive, and often times in analysis a quote could be considered part of multiple domains. As reflected in our paper, fear of testing can encompass both practical issues (fear of a positive result, fear of pain as a result of the nasal swab), and outcomes that may be a result of mistrust or even misinformation. Misinformation as a separate domain can refer to untrue beliefs that residents hold about COVID-19 and testing, that don’t necessarily precipitate fear, i.e., that COVID-19 is “fake.”

REVIEWER - Why were the responses from the predominantly Black group compared with the one Spanish speaking group, rather than with participants of other races in any group?

RESPONSE. Across all English-speaking focus groups, each group was predominantly or entirely comprised of non-Hispanic Black participants, so comparison of responses across other races is not feasible given the limited data. 

REVIEWER Discussion:

- "general lack of knowledge about testing recommendations, a similar lack of knowledge about risk factors for infection," - this was not presented at all in the Results

- The authors should discuss similar interventions that have been effective

RESPONSE: We have revised this sentence. 

REVIEWER - It would be helpful for the authors to suggest more concrete solutions to mitigating mistrust and increasing access (ie transportation, ensuring that tests are free, etc.)

RESPONSE: Our solutions were those pulled directly from FG participants, which are reported in the Results section. 

REVIEWER - The authors need a limitations paragraph, especially detailing the potential biases (ie voluntary response, potential underrepresentation/lack of representation of certain perspectives or subgroups), shortcomings of methods, limited generalizability to other populations, etc.

RESPONSE. This has been added to the discussion section. 

Sincerely,

Chigozirim Izeogu, MD MPH

---

## [Decision Letter · Decision Letter 1]

11 Nov 2022

PONE-D-21-34989R1Attitudes, perceptions, and preferences towards SARS Cov-2 testing and vaccination among African American and Hispanic public housing residents, New York City: 2020-2021PLOS ONE

Dear Dr. Gill,

Thank you for submitting your manuscript to PLOS ONE. After careful consideration, we feel that it has merit but does not fully meet PLOS ONE’s publication criteria as it currently stands. Therefore, we invite you to submit a revised version of the manuscript that addresses the points raised during the review process.

We look forward to receiving your revised manuscript.

Kind regards,

Sze Yan Liu, PhD

Academic Editor

PLOS ONE

Journal Requirements:

Additional Editor Comments :

As the reviewers note, this manuscript has been greatly improved. There are some minor revisions regarding additional information to include, primarily in the Introduction section, that would further strengthen this paper.

Reviewers' comments:

Reviewer's Responses to Questions

**Comments to the Author**

1. If the authors have adequately addressed your comments raised in a previous round of review and you feel that this manuscript is now acceptable for publication, you may indicate that here to bypass the “Comments to the Author” section, enter your conflict of interest statement in the “Confidential to Editor” section, and submit your "Accept" recommendation.

Reviewer #1: (No Response)

Reviewer #3: All comments have been addressed

2. Is the manuscript technically sound, and do the data support the conclusions?

Reviewer #1: Yes

Reviewer #3: Yes

3. Has the statistical analysis been performed appropriately and rigorously? 

Reviewer #1: Yes

Reviewer #3: N/A

4. Have the authors made all data underlying the findings in their manuscript fully available?

Reviewer #1: Yes

Reviewer #3: Yes

5. Is the manuscript presented in an intelligible fashion and written in standard English?

Reviewer #1: Yes

Reviewer #3: Yes

6. Review Comments to the Author

Reviewer #1: Thank you for making changes based on Reviewer recommendations. I believe the manuscript has been improved by your changes. I do have some additional relatively minor changes.

Introduction

Besides mentioning that there are 'social and structural determinants of health' impacting COVID-19 experiences between populations, is there any literature that you can mention in the Introduction on the beliefs or experiences related to testing and vaccination willingness/hesitancy? This may not be in African American and Hispanic groups, but I do believe the Introduction should acknowledge some of the research in this space, which if there is very little in these groups, leads to the gap this study is aiming to address.

Line 56 - this should be 'populations'.

Method

Line 102 - how were these neighborhoods and areas identified as having large infection rates?

Could you provide information on how the CSC fit in with the study – was it formed to conduct the study or was the study an offshoot of this wider approach?

Line 126 - correct extra I in COVIID.

Discussion

Line 296 - remove the 1 from this sentence - seems to be a typo.

The Discussion ends with a Limitations section. It would be a shame to end with this when the study has collected important data. I would recommend a short Conclusion summarising what was found and future research or public health initiatives that are suggested.

Reviewer #3: I appreciate the attention of the authors to addressing all my comments, and I feel that the manuscript has significantly improved. Just one minor thing remaining:

The authors state that they added in the Introduction that the vaccine became available in the middle of their research and therefore they added those questions to the focus group guide. However, I only see that mentioned in the Methods. It is important to add this to the Introduction.

Otherwise I am happy with the revisions made and recommend the manuscript for publication. No need to come back to me as long as the above addition to the Intro is made.

7. PLOS authors have the option to publish the peer review history of their article (what does this mean?). If published, this will include your full peer review and any attached files.

Reviewer #1: No

Reviewer #3: No

---

## [Author Response · Author response to Decision Letter 1]

26 Dec 2022

Dear Dr. Liu,

Thank you for reviewing our manuscript and for the opportunity to revise it to meet PLoS One’s publication standard. Please find listed below the comments submitted by the reviewers and our response to each comment in bolded italicized font. Please let us know if we can clarify anything further.

Reviewer #1: Thank you for making changes based on Reviewer recommendations. I believe the manuscript has been improved by your changes. I do have some additional relatively minor changes.

REVIEWER. Introduction. Besides mentioning that there are 'social and structural determinants of health' impacting COVID-19 experiences between populations, is there any literature that you can mention in the Introduction on the beliefs or experiences related to testing and vaccination willingness/hesitancy? This may not be in African American and Hispanic groups, but I do believe the Introduction should acknowledge some of the research in this space, which if there is very little in these groups, leads to the gap this study is aiming to address.

RESPONSE. We have added additional literature on SARS-CoV-2 testing hesitancy among Black adults in the United States to the Introduction, specifically lines 76-80. 

REVIEWER. Introduction. Line 56 - this should be 'populations'.

RESPONSE. We thank the reviewer for bringing this error to our attention and have edited our manuscript accordingly. 

REVIEWER. Methods. Line 102 - how were these neighborhoods and areas identified as having large infection rates?

RESPONSE. We have added a clarifying sentence to the Methods section to delineate how neighborhoods were selected for our study (lines 111-114). 

REVIEWER. Methods. Could you provide information on how the CSC fit in with the study – was it formed to conduct the study or was the study an offshoot of this wider approach?

RESPONSE. We have added additional details regarding the formation of the community steering committee, as well as a new citation directing readers to a recently published paper on the formation and function of the CSC (lines 133-135). 

REVIEWER. Methods. Line 126 - correct extra I in COVIID.

RESPONSE. We thank the reviewer for bringing this error to our attention and have edited our manuscript accordingly.

REVIEWER. Discussion. Line 296 - remove the 1 from this sentence - seems to be a typo.

RESPONSE. We thank the reviewer for bringing this error to our attention and have edited our manuscript accordingly.

REVIEWER. Discussion. The Discussion ends with a Limitations section. It would be a shame to end with this when the study has collected important data. I would recommend a short Conclusion summarising what was found and future research or public health initiatives that are suggested.

RESPONSE. We have added an additional paragraph to the end of our Discussion section (lines 405-418). 

Reviewer #3: I appreciate the attention of the authors to addressing all my comments, and I feel that the manuscript has significantly improved. Just one minor thing remaining:

REVIEWER. Introduction. The authors state that they added in the Introduction that the vaccine became available in the middle of their research and therefore they added those questions to the focus group guide. However, I only see that mentioned in the Methods. It is important to add this to the Introduction. Otherwise I am happy with the revisions made and recommend the manuscript for publication. No need to come back to me as long as the above addition to the Intro is made.

RESPONSE. We have added this to our revised manuscript (lines 105-108).

---

## [Decision Letter · Decision Letter 2]

3 Jan 2023

Attitudes, perceptions, and preferences towards SARS Cov-2 testing and vaccination among African American and Hispanic public housing residents, New York City: 2020-2021

PONE-D-21-34989R2

Dear Dr. Gill,

We’re pleased to inform you that your manuscript has been judged scientifically suitable for publication and will be formally accepted for publication once it meets all outstanding technical requirements.

Kind regards,

Sze Yan Liu, PhD

Academic Editor

PLOS ONE

Additional Editor Comments (optional):

Reviewers' comments:

Reviewer's Responses to Questions

**Comments to the Author**

1. If the authors have adequately addressed your comments raised in a previous round of review and you feel that this manuscript is now acceptable for publication, you may indicate that here to bypass the “Comments to the Author” section, enter your conflict of interest statement in the “Confidential to Editor” section, and submit your "Accept" recommendation.

Reviewer #1: All comments have been addressed

2. Is the manuscript technically sound, and do the data support the conclusions?

Reviewer #1: Yes

3. Has the statistical analysis been performed appropriately and rigorously? 

Reviewer #1: Yes

4. Have the authors made all data underlying the findings in their manuscript fully available?

Reviewer #1: Yes

5. Is the manuscript presented in an intelligible fashion and written in standard English?

Reviewer #1: Yes

6. Review Comments to the Author

Reviewer #1: (No Response)

7. PLOS authors have the option to publish the peer review history of their article (what does this mean?). If published, this will include your full peer review and any attached files.

Reviewer #1: No

---

## [Editor Report · Acceptance letter]

9 Jan 2023

PONE-D-21-34989R2 

Attitudes, perceptions, and preferences towards SARS CoV-2 testing and vaccination among African American and Hispanic public housing residents, New York City: 2020-2021 

Dear Dr. Gill:

I'm pleased to inform you that your manuscript has been deemed suitable for publication in PLOS ONE. Congratulations! Your manuscript is now with our production department. 

Kind regards, 

on behalf of

Dr. Sze Yan Liu 

Academic Editor

PLOS ONE